# Investigating the Biological Potency of Nitazoxanide-Based Cu(II), Ni(II) and Zn(II) Complexes Synthesis, Characterization and Anti-COVID-19, Antioxidant, Antibacterial and Anticancer Activities

**DOI:** 10.3390/molecules28166126

**Published:** 2023-08-18

**Authors:** Abeer A. Sharfalddin, Inas M. Al-Younis, Abdul-Hamid Emwas, Mariusz Jaremko

**Affiliations:** 1Department of Chemistry, Faculty of Science, King Abdulaziz University, P.O. Box 80203, Jeddah 21589, Saudi Arabia; 2Biological and Environmental Sciences and Engineering Division, King Abdullah University of Science and Technology (KAUST), Thuwal 23955-6900, Saudi Arabia; inas.younis@kaust.edu.sa; 3Core Labs, King Abdullah University of Science and Technology (KAUST), Thuwal 23955-6900, Saudi Arabia; abdelhamid.emwas@kaust.edu.sa; 4Smart-Health Initiative (SHI) and Red Sea Research Center (RSRC), Division of Biological and Environmental Sciences and Engineering (BESE), King Abdullah University of Science and Technology (KAUST), Thuwal 23955-6900, Saudi Arabia

**Keywords:** nitazoxanide, transition metals, anticancer, anticancer, COVID-19, antioxidant, antibacterial, DFT calculation

## Abstract

In this work, the biological potency of nitazoxanide (NTZ) was enhanced through coordination with transition metal ions Cu(II), Ni(II), and Zn(II). Initially, complexes with a ligand-metal stoichiometry of 2:1 were successfully synthesized and characterized by spectroscopic techniques and thermogravimetric methods. Measurement of the infrared spectrum revealed the bidentate nature of the ligand and excluded the possibility of the metal ion—amide group interaction. Nuclear magnetic resonance spectra showed a reduction in the NH- intensity signal and integration, indicating the possibility of enolization and the formation of keto-enol tautomers. To interpret these results, density functional theory was utilized under B3LYP/6-311G** for the free ligand and B3LYP/LANL2DZ for the metal complexes. We used UV-Vis and fluorescence spectroscopy to understand the biological properties of the complexes. This showed stronger interactions of NTZ-Cu(II) and NTZ-Ni(II) with DNA molecules than the NTZ-Zn(II) compound, with a binding constant (K_b_) for the copper complex of 7.00 × 10^5^ M^−1^. Both Cu(II)- and Ni(II)-NTZ had functional binding to the SARS-CoV-2 (6LU7) protease. Moreover, all metal complexes showed better antioxidation properties than the free ligand, with NTZ-Ni(II) having the best IC_50_ value of 53.45 μg/mL. NTZ-Ni(II) was an effective antibacterial, with a mean inhibitory concentration of 6 μM, which is close to that of ampicillin (a reference drug). The metal complexes had moderated anticancer potencies, with NTZ-Cu(II) having IC_50_ values of 24.5 and 21.5 against human breast cancer cells (MCF-7) and cancerous cervical tumor cells (HeLa), respectively. All obtained complexes exhibited high selectivity. Finally, the metal ions showed a practical role in improving the biological effectiveness of NTZ molecules.

## 1. Introduction

Drug discovery processes annually lead to new chemical entities that have promising activity against particular biological targets. Some of these entities have serious toxicity, while others have low cell penetration due to poor lipophilicity and solubility. Strategies to overcome these complications include drug encapsulation in nanocarriers, the development of prodrugs, and structural modifications by the addition of active groups or heterocycles. Additionally, the coordination of biologically active molecules with metal centers may improve the pharmacological profile with reduced drug resistance, lowered toxicity, and increased absorbance and stability in biological environments [1]. Moreover, metal ions can improve the intracellular accumulation of the drug, alter absorption into the bloodstream, and increase the number of clinical applications. For instance, copper ions enhance the antidiabetic, antimalarial, and anticancer properties of metformin (a drug for type 2 diabetes) [2,3]. The anticancer potency of metformin was investigated by observing binding to copper; this metal ion is essential for the epithelial-to-mesenchymal transition (EMT). The formation of a stable Cu-metformin compound leads to the destruction of the EMT and reduces tumor stemness [3]. Another example of metal-promoting ligand biological properties is the zinc sulfadiazine complex in a 1:2 metal-to-ligand molar ratio. This zinc sulfadiazine complex has been used to promote wound healing and control infections [4]. Additionally, Ag(I) has been reported to improve the antimicrobial efficiency of sulfadiazine for the treatment of first, second, and third-degree burns [5,6].

Nitazoxanide (NTZ) is a nitrothiazolyl-salicylamide compound that has antimicrobial [7], antiviral [8], antiprotozoal [9], and anticancer effects [10,11]. The molecular structure has different donating atoms, including ester oxygen, an amide group, sulfur, and nitrogen in the thiazole ring. These molecules serve as excellent bidentate ligands and easily form complexes via the amide oxygen and thiazole nitrogen with Pt(II), Pt(II), and Ru(III) [12], whereas Ni(II) and Co(II) ions coordinate with the thiazole nitrogen at a molar ratio of 1:1 [13]. Among these mentioned complexes, [Ni(NTZ)(CH_3_COO)(OH_2_)]·CH_3_COO had improved toxicity against Gram-positive Staphylococcus aureus. Our previous work describing the coordination of NTZ with the novel metals Ru(III), Au(III), and Ag (I) demonstrated increased biological potency of the organic ligand when combined with the metal ion [14]. Both Ru(III) and Au(III) complexes showed practical antioxidant and anticancer activities, but the Ag (I) complex showed the greatest inhibition of Gram-negative *Escherichia coli*.

Another study described the interaction of NTZ with the divalent transition metals Mn(II), Cu(II), and Zn(II) [15] and with Ag(I) [16]. Unfortunately, these spectroscopy studies did not examine accompanying biological changes occurring in the modified NTZ molecule.

Among the transition metal ions, zinc, copper, and nickel are essential components for normal biological functioning and have essential roles in the human body [17]. For example, activation centers catalyze numerous enzymes process, energy metabolism and central nervous system activity. 

Therefore, in this study, three NTZ-based Cu(II), Ni(II), and Zn(II) compounds were synthesized and characterized using different spectroscopy approaches: Fourier transform infrared (FT-IR), electron paramagnetic resonance (EPR), nuclear magnetic resonance (NMR), and UV−visible spectroscopy (UV-Vis) along with thermogravimetric analysis (TGA). A density functional theory (DFT) approach was used to estimate the chemical parameters and chemical reactivity of the synthesized compounds. Biological properties were investigated by observing the interactions with DNA and HSA using UV-Vis and fluorescence spectroscopy titration. In addition, the antibacterial activity against *Escherichia coli* and the cytotoxicity against two human cancer cell lines (MCF-7 and HELA) and normal cells were measured in vitro. Finally, the ABTS*+ radical-scavenging assay and molecular docking were used to examine the antioxidant activity and antiviral efficiency against SARS-CoV-2 (6LU7), respectively, Figure 1. 

## 2. Results and Discussion

### 2.1. General Chemical and Physical Characterization

The metal complexes were synthesized, and the complexes’ structures were identified by several spectroscopy approaches (^1^H-NMR, FTIR, UV–visible, EPR) and TGA. The metal complexes are stable at normal atmospheric conditions and easily soluble in DMF and DMSO only. The obtained molar ratio suggested a stoichiometric ratio of 2:1 (ligand to metal) for all metal complexes Appendix A. The molar conductance was <50 Ω^−1^ cm^2^ mol^−1^, suggesting the three complexes are non-electrolytic [6].

To find the coordination mode of the obtained complexes, a careful comparison was performed between the free ligand and its complexes (Figure 1). The NTZ IR spectrum has stretching bands at 3360, 1770, 1660, and 1600 cm^−1^ assigned to v(NH), υ(C=O)^ester^, v(C=O)^amide^, and v(C=N)^thiazole^, respectively. The stretching mode of NH was maintained in both the Cu(II) and Ni(II) complexes. However, it was shifted to higher wavenumbers upon chelation with Zn(II), which excluded the association of the NH- group in the reaction. The sharp peak for v(C=O)^ester^ (1770 cm^−1^), however, showed no change. The bidentate nature of the ligand was evident by the shifts in the vibrational frequencies of v(C=O)^amide^ and v(C=N)^thiazole^ to lower frequencies, with a lower intensity that overlapped with the v(C=C) vibrations. These results correlated with the calculated frequency values, and the spectra are presented in Appendix A. 

^1^H NMR is a powerful technique for investigating the coordination structure of the obtained complexes in solution media. The spectra were recorded in DMSO-d_6_ and are presented in Figure 2. The spectrum of the NTZ-Zn(II) compound showed the same signals as the original ligand: singlet signals at 13.60, 8.86, and 2.20 ppm, which correspond to amide (–NH), thiazole (–CH), and methyl (–CH_3_) groups; the phenyl ring presented two doublet and triblet signals between 7.20–7.79 ppm. The NH- signal showed a decrease in intensity and integration. However, the IR data recorded in the solid state excludes the interaction of the metal ion with the amide group (NH-), which suggested an enolization process and formation of a keto-enol tautomer occurred (Figure 3). It has been reported that the coordination of metal ions can compete with the keto form of the species and stabilize enol metal complexes [18]. To investigate this possibility, a DFT calculation was carried out to assess the most stable structure of the two tautomers using B3LYP/6-311G** for the free ligand and B3LYP/LANL2DZ for the metal complex. 

A gas phase calculation showed an equilibrium between the keto and enol forms of NTZ with a ΔE (energy difference of the keto and enol forms) of 28.76 kJ/mol. For the NTZ-Zn(II) compound, the obtained value of the keto isomer was higher than for the enol structure in a vacuum atmosphere (Table 1). The effect of the solvent (DMSO) on the compound’s predominant form was investigated. The solvent had a negligible effect on the keto-enol equilibrium, indicating that the keto structure competes with the more stable enol isomer. 

UV-VIS electronic absorption spectra of 2 × 10^−2^ M NTZ in DMSO showed two bands at λ_max_ = 275 and 345 nm. This was attributed to the excitation of the π-electron of the aromatic system to π* [19] and a n→π* transition containing carbonyl lone pair electrons. Upon complexation, these bands showed a red shift, indicating interaction with the metal ion. The low-intensity band at λ_max_ = 420 nm was overlapping and appeared with high intensity at 452, 441 and 447 nm for Cu(II), Zn(II), and Ni(II), respectively. This could be assigned to a metal-to-ligand charge transfer (MLCT) [12] (Appendix A). The Cu(II) complex showed absorption at 790 nm, which was assigned to the transition state ^2^A_1_→^2^E in the octahedral geometry, while the Ni(II) complex showed two low-intensity bands at 670 and 755 nm, which revealed octahedral geometry of [Ni(NAT)_2_Cl_2_]. The electron configuration of the Zn(II) complex is d^10^; hence, no bands were observed in the range of 500–800 nm due to the diamagnetic property of the metal. 

EPR spectroscopy was employed to study the coordination of the Cu(NTZ)_2_Cl_2_ complex in DMSO at room temperature, as described previously [20]. Figure 4 shows two peaks, one with parallel orientation (gǁ) and the other with perpendicular orientation (g⊥). The obtained g values revealed that gǁ (2.14) was higher than g⊥ (2.29) and suggested compressed Z axial symmetry and elongation in the equatorial plane (*x*- and *y*-axes) [21,22]. The octahedral arrangement has oxygen and nitrogen atoms in the equatorial plane and Cl atoms in the axial position. The unpaired electrons for the Cu(II) ground state are ^2^A_1g_ (d_z2_). The relative equation g_av_ = (g‖ + 2 g_┴_)/3 was used to study the nature of binding to the NTZ ligand, and the calculated value was 2.24 < 2.3, which indicates the metal–ligand bond is between ionic and covalent in character [23].

TGA is a practical approach to studying the thermal stability and coordinated water molecules and confirming the obtained structure [6]. Appendix A presents the thermal decomposition curve of the isolated metal complexes in the range of 25–800 °C. Generally, all complexes showed good stability up to 200 °C with two pyrolysis stages ending with the metal oxide as a metallic residue. Both Cu(II) and Ni(II) complexes started with one exothermic peak, which indicates the loss of one anhydrase water molecule, two chloride ions, a carbon oxide, and a nitro group at the first step with a mass loss of 28.8% and 29.9%, respectively. The second step, which showed an endothermic peak, brought the total mass loss to 76.9% and 78.5%, which resembled the calculated values of 78.0% and 79.9%. The Zn(II) complex did not present hydrated water or solvent molecules at the first stage and showed evaporated solvents and moisturized water before 200 °C. In the first step, it released two chloride ions, C_2_H_6_, carbon oxide, and a nitro group, while the rest of the molecule degraded slowly from 255–770 °C. The decomposed stages agreed with the complex formulas in Table 2.

### 2.2. Computational Studies 

Different attempts using various solvents with slow cooling methods failed to produce single crystals of the metal complexes. Thus, the spectroscopic and thermal investigation results were used to build structures of the ligand coordinated with the metal with the Gaussian program. The results showed that all compounds may have a distorted octahedral geometry (Figure 5). There was a practical change in the formed bond length and angles after the coordination with the transition metals. In detail, the M–O and M–N bonds between the metal and free ligand atoms were in the range of 2–2.3 Å and showed small and medium ionic character, respectively [24]. The formed Ni-N had a bond length of 1.90 Å and showed ionic character [17]. Complex angles around the respective central metal ion indicated that electric repulsion between the coordinated atoms distorted the structure and stabilized the molecules. Moreover, because the electron configuration of Cu(II) is d^9^, the extra stability of the copper complex is mainly due to the Jahn–Teller effect [25]. Appendix A illustrates the selected geometric bond lengths and bond angles around the metal ion of the optimized complexes. There was negligible change in the bond and angles in the other ligand atoms after the coordination reaction. 

An activity rank of the complexes was determined using quantum chemical descriptors such as the highest occupied molecular orbital energy (E_HOMO_), lowest unoccupied molecular orbital (E_LUMO_), the energy gap (∆E), absolute electronegativities (χ), absolute hardness (η), global softness (S), chemical potential (Pi), and global electrophilicity (ω) [26,27]. Frontier molecular orbital (FMO) maps were generated via the HOMO and LUMO orbitals (Figure 6). Generally, the HOMO orbital is located around the metal ion for the obtained complexes, while the LUMO orbital is distributed over most of the ligand molecules in the NTZ-Ni(II) complex and just over one coordinated ligand in the NTZ-Cu(II) and NTZ-Zn(II) complexes. One good indicator of the molecule reactivity pattern is ∆ E [26], with calculated values presented in Table 3. A small ∆E indicates high chemical reactivity; thus, NTZ-Cu(II) has the best reactivity among the metal complexes, followed by NTZ-Zn(II) and then NTZ-Ni(II). Calculated chemical hardness (η) and chemical softness (S) values showed that NTZ-Ni(II) is a hard complex and has high stability, whereas the other complexes are chemically soft. 

Moreover, in Table 3, electrophilicity (ω) describes a compound’s ability to obtain an additional electronic charge and is an indicator of reactivity during biological effects, including inhibition. Based on this data, biological activity can be ordered as NTZ-Cu(II) > NTZ-Ni(II) > NTZ-Zn(II), meaning the copper compound has favorable toxicity and inhibitory activity [28]. As for the additional electronic charge (ΔN_max_) in Table 3, it indicates the maximum charge that an electrophile may accept from the environment [29]: NTZ-Ni(II) had the highest value. Finally, all-inclusive processes for the metal complexes are spontaneous due to the negativity of the chemical potential (Pi). The obtained quantum descriptors show that NTZ-Cu(II) and NTZ-Ni(II) have practical chemical reactivity. 

### 2.3. Biological Activities 

#### 2.3.1. Absorption and Fluorescence Studies with Protein (HSA)

Interactions of the synthesized metal complexes with HSA were measured using absorption titration and fluorescence quenching experiments at physiological conditions. Alterations of the HSA absorption spectra after adding different concentrations of NTZ metal complexes (0 to 1.9 μL) are shown in Figure 7 and Appendix A. The results indicate a hyperchromic mode at 280 nm with a red shift of ~3 nm and increasing intensity of the 434 band with increasing metal complex concentration. These observations are attributed to the destruction of the tertiary structure of HSA and the extension of additional aromatic acid residues into the aqueous environment [30]. The calculated intrinsic binding constants (K_b_) were 4.79 × 10^4^, 7.46 × 10^4^ and 1.87 × 10^5^ M^−1^ for NTZ-Ni(II), NTZ-Zn(II) and NTZ-Cu(II), respectively. The binding of copper was found to introduce a conformational change in the HSA structure.

The fluorescence spectra of the HSA protein are given in Figure 8 and Appendix A. These show that the intensity of the emission at 349 nm was quenched during the gradual addition of the metal complexes. This effect resulted from the interaction of the indole moiety of the Trp chromophore in the HSA with the metal complexes [26]. Moreover, the wavelength of the emission maximum of HSA changed, attributed to a practical modification of the cavity polarity around the Trp residue [31]. The quenching constant (K_SV_) was calculated by the Stern-Volmer equation and found to be 5.6 × 10^6^, 4.6 × 10^6^ and 4 × 10^6^ M^−1^ for NTZ-Cu(II), NTZ-Ni(II) and NTZ-Zn(II), respectively. This revealed a possible interaction of the copper complex with HSA. 

#### 2.3.2. Absorption DNA Binding Studies

The UV–visible absorption titration method was used to study the interaction of Ct-DNA to the metal complexes (Figure 9 and Appendix A). The absorption peaks at 275, 355, and 440 nm showed a small red shift of 1–3 nm upon gradually increasing Ct-DNA concentration. Both the Cu(II) and Ni(II) complexes showed a decrease in absorption intensity, while Zn(II) showed an increase in absorption intensity. The decreasing absorption intensity suggests structural modification of the target DNA after adding the metal complexes [32]. Furthermore, external contact or a partial effect of the metal ions unraveled the helical structure of DNA [33]. However, the hyperchromic absorptivity of the Zn(II) complex indicated weak intercalation to the base pairs of DNA [34]. 

The computed binding constants (K_b_) were 4.14 × 10^5^, 5.78 × 10^5^ and 7.00 × 10^5^ M^−1^ for Zn(II), Ni(II) and Cu(II) complexes, respectively. These values reflect an intercalator binding type in the range of the ruthenium complex [32], with the copper complex having the strongest binding.

#### 2.3.3. In Silico Study of NTZ Complexes COVID-19 Binding

A number of reported clinical trials for COVID-19 have suggested NTZ is a highly promising drug alone or in combination with other antivirals [35]. Compared with other antiviral drugs (favipiravir, remdesivir, umifenovir, and hydroxychloroquine), NTZ can act in various phases of the disease, including the severe phase, due to its ability to reduce cytokine storms [36]. A previous procedure was utilized [37,38] to dock the metal compounds into the active site in the SARS-CoV-2 protease (6LU7) to investigate their inhibitory effects against COVID-19. One of the active sites in this protease is Cys145 [36], where cysteine thiol is an electron-donating group and plays a major role in the proteolytic cleavage process [39]. Free ligand NTZ showed a molecular docking score of −6.39776 KJ/mol and revealed higher levels of binding to the protein with high docking accuracy (RMSD). Cys145 showed a back donor property to carbonyl oxygen in the molecule, while the thiazole ring formed an arene-H bond with Gln 189 (Table 4). Introducing metal ions improves the biological activities of NTZ as an antiviral agent. Both Cu(II) and Ni(II) NTZ complexes presented a practical enhancement, with docking scores of −8.14215 and −8.73124 KJ/mol, respectively. The RMSD for NTZ-Cu(II) was around 1.7 Å, indicating a better docking pose. On the other hand, the zinc complex did not improve the antiviral potential, with a binding energy of −5.62794 KJ/mol. The interaction of the Ni(II) complex is illustrated in Table 4 in 2D and 3D and presents the three main hydrogen bonds between MET165 and CYS145. The copper complex showed more interactions, and chloride atoms acted as H-acceptors with GLN189, MET165, and GLU166. The fourth bond was donated by the NH hydrogen to MET49. These results reveal how metal ions enhance antiviral potency. The ranking energies with RMSD of the repurposed complexes against the 6LU7 protein are shown in Appendix A.

#### 2.3.4. Antioxidant Assay

The presence of a salicylamide moiety and nitrothiazole in NTZ suggests possible antioxidant activity [36]. Therefore, the ABTS assay was used to measure the IC_50_ values of the NTZ ligand and the metal complexes (Table 5). IC_50_ values ranging from 50 to 100 μg/mL reveal intermediate antioxidant activity [40]. Initially, metal complexes showed better radical scavenging than the free ligands, due to the metal ions role, which allow reactions with free radicals to occur faster [41]. While the investigated NTZ-Cu(II), NTZ-Ni(II), and NTZ-Zn(II) complexes had intermediate antioxidant activity, NTZ-Ni(II) had stronger antioxidant activity, which may be sufficient to remove the free radicals and other reactive oxygen species (ROS) that are at times found elevated in cancer, diabetes, and neurodegenerative diseases [41]. 

#### 2.3.5. Antibacterial Activity

The antibacterial properties of each complex were studied using *Escherichia coli*. This bacterium was chosen due to it causing serious foodborne illnesses and being easy and cheap to handle in scientific experiments [42]. Measured zones of inhibition ranged from 8–26 mm (Table 6), demonstrating that the metal complexes generally outperformed the free ligand. Among the metal complexes, NTZ-Ni(II) showed the highest antibacterial activity (Figure 10 and Appendix A). This could be attributed to the high electrophilicity of the compound as the extracted theoretical calculation presented above.

#### 2.3.6. In Vitro Antiproliferative Activity

Di Santo et al. suggested that NTZ can be utilized against cancer due to its ability to target tumor cells and obstruct cell interactions with the microenvironment [11]. To investigate this potential, we examined the effects on two cancer cell lines: MCF-7, a human breast cancer cell line, and HeLa, a cancerous cervical tumor cell line. HEK293, a human embryonic kidney cell line, was used as a control to understand the selectivity and safety of the compounds. Table 7 compares NTZ with cisplatin, a standard anti-cancer drug for secondary breast cancer [43]. Higher IC_50_ values were found upon complexation with metal ions, reflecting the cytotoxicity of the metal ions. The collected values of IC_50_ for the metal complexes were in the moderate range toward both cell lines. The NTZ-Cu(II) showed effective inhibition toward HeLa (IC_50_ = 70 µM ) and then MCF-7 (IC_50_ = 78.19 µM )cell lines. Moreover, all investigated complexes have good selectivity toward the normal cell line HEK cells better than the free ligand.

## 3. Experimental

### 3.1. Chemicals and Instruments

NTZ with a purity of 99% (C_12_H_9_N_3_O_5_S) was purchased from Sigma Aldrich. The chloride metal salts were of analytical grade and used without further purification. The biological experiments were carried out using human serum albumin (HSA, A1887; globulin and fatty acid-free) and calf thymus DNA (Sigma Aldrich, Burlington, MA, USA). 

The molar conductivity of a solution containing a complex of concentration 10^−3^ M in N,N-dimethylformamide (DMF) was measured using a HACH conductivity meter. A Bruker 600 MHz spectrometer was used to record the ^1^H-NMR spectra in DMSO d_6_ solution. Infrared (IR) spectra for the ligand and metal complexes were recorded on a Bruker IR spectrophotometer in the range of 400–4000 cm^−1^, and the uncertainty did not exceed 0.5 nm at room temperature. A Shimadzu UV/Vis spectrometer was used to collect the electronic spectra of the metal complexes in the range 200–800 nm, while fluorescence experiments were carried out on a Cary Eclipse spectrofluorometer from 300 to 600 nm. A NETZSCH STA 449F1 thermal analysis system was used for the TG-DSC experiments with an airflow rate of 30 mL/min, a heating rate of 10 °C/min, and a temperature range of 25–800 °C; the data were analyzed using Proteus software. The percentage of metal ions was calculated thermogravimetrically as metal oxides. Electron paramagnetic resonance (EPR) was recorded on a Bruker EMX PLUS spectrometer using the X band frequency (9.5 GHz) in DMSO solution and room temperature.

### 3.2. Synthesis of Metal Complexes

The solubility of NTZ in water, ethanol, and acetone was very low [45,46]. Thus, 20% DMF was added to the ethanol solution [46]. An aqueous metal solution of NiCl_2_ (0.129 mg/15 mL), CuCl_2_ (0.17 mg/15 mL), or ZnCl_2_ (0.172 mg/15 mL) was added dropwise to ligand solution with stirring to a final volume of 30 mL. The mixtures were refluxed for 2 h. Then, the colored precipitates were collected, washed, and dried in a desiccator. 

**[Cu(NTZ)_2_Cl_2_]:** color, dark green. IR (FT-IR, cm^−1^): 3360 (NH), 1770 (C=O), 1664 (NH-C=O), 1600 (N-C), 1384, 1292 (NO_2_), 1170 (C-O), 874, 651. UV-Vis (DMSO, 10^−3^, nm): 279, 349, 452, 790. Molar conductance (10^−3^ M, DMSO, Ω^−1^ cm^2^ mol^−1^): 7.45.

**[Ni(NTZ)_2_ Cl_2_]:** color, light green. IR (FT-IR, cm^−1^): 3360 (NH), 1775 (C=O), 1664 (NH-C=O), 1600 (N-C), 1384, 1292 (NO_2_), 1153 (C-O), 870, 650. UV-Vis (DMSO, 10^−3^, nm): 266, 342, 447, 670 and 755. Molar conductance (10^−3^ M, DMSO, Ω^−1^ cm^2^ mol^−1^): 24.

**[Zn(NTZ)_2_ Cl_2_]:** color, beige. IR (FT-IR, cm^−1^): 3430 (NH), 1770 (C=O), 1669 (NH-C=O), 1600 (N-C), 1384, 1292 (NO_2_), 1177 (C-O), 871, 650. ^1^H NMR (DMSO-d_6_, 500 MHz, ppm): 13.60 (b, 1H, NH), 8.86 (s, 1H, thiazole), 7.82 (dd, 1H, CHAr), 7.70 (td, 1H, CHAr), 7.47 (td, 1H, CHAr), 7.32 (dd, 1H, CHAr), 2.20 (s, 3H, CH_3_). UV-Vis (DMSO, 10^−3^, nm): 280, 346 and 441. Molar conductance (10^−3^ M, DMSO, Ω^−1^ cm^2^ mol^−1^): 40.5.

### 3.3. Theoretical Calculations

#### 3.3.1. Optimization Procedure 

The optimization of the obtained novel metal complexes was carried out using Gaussian 09 software version6 [47] to identify the most stable electronic structures of all compounds. The B3LYP hybrid functional-based DFT method was used, with the triple-zeta 6–311+G(d,p) basis set for the calculation of the free ligand. The correlation-consistent LANL2DZ basis set with 6–311+G(d,p) was employed as mixed basis sets for the calculation of the Cu(II), Ni(II), and Zn(II) ions [48] which the first was implemented to the metal atoms while the second for all atoms. Gauss View software was used to build the input files and to visualize the output files. The harmonic frequencies were scaled by a factor of 0.966 and 0.961 for the 6-31G(d,p) and LANL2DZ basis sets, respectively [44]. The structures were on the potential energy surface and confirmed by the absence of an imaginary frequency. The HOMO and LUMO energies and the orbital levels were extracted using Gauss View to calculate the essential quantum parameters, including the energy gap (E_gap_ = E_LUMO_ − E_HOMO_), absolute electronegativity (χ = −E_HOMO_ +E_LUMO_/2), absolute hardness (ɳ = E_LUMO_ − E_HOMO_/2), chemical potential (μ = −χ), global softness (S = 1/2ɳ), and global electrophilicity (ω = μ^2^/ɳ)) [26]. 

#### 3.3.2. Molecular Docking Platforms

The assessment of free NTZ and metal complexes as antiviral agents was done using Molecular Operating Environment (MOE) software against the crystal structure of SARS-CoV-2 protease (PDB ID: 6LU7). The PDF file was obtained from the RCSB Protein Data Bank (PDB). The initial step in the preparation process was to remove any redundant chains, water molecules, and co-ligands from the SARS-CoV-2 protease structure. The distribution of partial charges was then corrected using the MOE-Quick prep feature. The active sites were located with the MOE-Site Finder tool and expressed as dummy atoms. The optimized geometries of the metal complexes were converted to MDB files and saved as a database. The docking protocol was carried under the default option; the triangle matcher method was used to place the compound, and GBVI/WSA dG and London dG were used for rescoring and scoring, respectively. The results were presented in five different poses, each with a corresponding binding energy measured in units of kcal/mol. The first pose was selected for further analysis.

### 3.4. In Vitro Binding of Complexes to Biological Molecules

Spectroscopic titration experiments were performed by adding a constant concentration of HSA to the samples. First, the HSA solution concentration was determined by the absorption spectra taken at 278 nm and extinction coefficient of 35,219 M^−1^cm^−1^ using quartz cuvettes of 1 cm path length. The absorption spectral method and the binding constant calculations are explained in our previous work [17]. For the fluorescence technique, the emission wavelengths were set at 300–400 nm. Increasing concentrations of the metal complex in the range of 0.07 to 0.25 M were added to a constant concentration of HSA, and the fluorescence emission was subsequently measured. The Stern–Volmer equation was used to calculate K_SV_: F_o_/F (or I_o_/I) = 1 + KSV [HSA], where F_o_ is the fluorescence intensity in the absence of quencher, and F is the fluorescence intensity in the presence of metal complexes. K_SV_ is the Stern–Volmer quenching constant and was obtained from the slope of the F_o_/F plot of [Complex] [26].

UV-Vis was used to study the binding of metal complexes to calf thymus-DNA. The experiments used a constant compound concentration and gradually increased Ct-DNA concentrations from 1.57 to 5.15 μM. CT-DNA was dissolved in Tris-HCl buffer solution (pH 7.4) and confirmed to be free of protein using a UV absorbance ratio (A_260_/A_280_) of 1.8–1.9 [26,28]. Effective binding constants (K_b_) were evaluated using the Benesi-Hildebrand equation.
(1)$$\frac{A_{o}}{A-A_{o}}=\frac{{Ԑ}_{G}}{{Ԑ}_{H^{-}G}-{Ԑ}_{G}}+\frac{{Ԑ}_{G}}{{Ԑ}_{H^{-}G}-{Ԑ}_{G}}\times \frac{1}{K\left[\mathrm{DNA}\right]}$$
where A_o_ and A are the absorbances of the drug and its complex with DNA, respectively, and ${Ԑ}_{G}$ and ${Ԑ}_{H^{-}G}$ are the absorption coefficients of the drug and drug–DNA complex, respectively. 

### 3.5. Assay for Antioxidant Properties

The ABTS*+ radical-scavenging activity of the produced complexes was evaluated following Re et al. [49]. A 7 mM solution of 2,2′-Azinobis-[3-ethylbenzthiazoline-6-sulfonic acid] (ABTS) was created by dissolving ABTS twice in distilled water. The ABTS solution was combined with K_2_S_2_O_8_ solution at a final concentration of 2.45 mM and left to react for 12 to 16 h at room temperature in the dark. The ABTS solution was diluted with a suitable solvent (methanol) to an absorbance of 0.7 ± 0.01 to assess the antioxidant capacity of the produced complexes in comparison to the standard or reference (ascorbic acid). In a 96-well plate, 190 µL of ABTS reagent was mixed with 10 µL of the standard or samples. After a five-minute incubation in the dark, the plate was shaken for 10 s at medium speed, and the absorbance was recorded at 734 nm. Ascorbic acid was used as the reference for the evaluation of the radical-scavenging activity of the produced complexes at various concentrations (1, 3, 5, and 10 µg/mL). At least three measurements were taken for each condition. To compute the absorbance suppression rate, the ABTS+ scavenging effect (%) = ((AB − AA)/AB) × 100 was used, where AB is the absorbance of the ABTS radical + methanol, and AA is the absorbance of the ABTS radical + the manufactured complex/standard. 

### 3.6. Antibacterial Activity

All synthesized compounds were tested for their antibacterial efficacy against pathogenic strains using the agar disc diffusion and broth dilution methods. The Gram-negative bacterial pathogen employed in this study was *Escherichia coli* (ATCC: 25922) [50]. On a Muller-Hinton agar (MHA) plate, the bacterial strain was cultured and incubated for 18–24 h at 35 °C. 20 mg of each substance under examination was produced; DMSO also served as a negative control. Ampicillin, a standard drug with strong antibacterial properties, was utilized. Using a sterile swab and a bacterial culture that was adjusted to McFarland 0.5 standard solution (1.5 × 10^8^ CFU/mL), Hinton agar plates were lawned. Individual paper discs 5 mm in diameter were impregnated with the chemicals at a fixed concentration of 100 g/mL. The antibacterial activity of each test sample was evaluated twice using the minimal inhibitory concentration (MIC) method, measuring the diameter of the inhibition zone and comparing it with a drug standard. For the DMSO sample, no inhibitory zone was observed.

### 3.7. Cytotoxicity Assessment (MTT Assay)

Three cell lines were selected to test anticancer activity. The first was HeLa, which are malignant cervical tumor cells. The second was MCF-7, human breast cancer cells. The third was HEK293 (human embryonic kidney cells), a normal cell line. Each cell line was cultured in Dulbecco’s modified eagle’s medium (DMEM) with 0.2% sodium bicarbonate and 10% fetal bovine serum (FBS). The cells were kept at 37 °C in 5% CO_2_ and 95% humidity. Cell vitality was evaluated using the MTT test [51]. MTT tetrazolium dye is reduced by enzyme activity, measuring cell number and health. Batches of cells with more than 98% cell viability were employed. In 96-well culture plates, 10,000 cells were placed and left for 24 h to adhere in a CO_2_ incubator set at 37 °C. MTT (5 mg/mL stock in PBS) was added after drug exposure (10 μL/well of 100 μL cell suspension), and plates were then incubated for an additional 4 h in the CO_2_ incubator. After discarding the supernatant, 200 μL DMSO was added to each well, and the wells were gently mixed. An Evolution 201 Reader was used to read the produced color at 550 nm (Thermo Scientific, Waltham, MA, USA).

The same conditions were used for the untreated control sets as well. Each measurement was adjusted for background absorbance. Following treatment with the synthetic compounds, survival curves for each cell line were created by graphing the relationship between the number of cells that survived and the amount of compound. The 50% inhibitory concentration (IC_50_) was calculated from charts of the dosage response curve for all substances using Prism software 6 (San Diego, CA, USA).

## 4. Conclusions

Here, we show that the biological properties of NTZ are enhanced by the three metal ions Cu(II), Ni(II), and Zn(II). The synthesized complexes were characterized by different spectroscopic techniques and TGA. The NMR spectrum of the NTZ-Zn(II) compound showed a reduction of the NH- intensity signal and integration, suggesting the NH contribution in the coordination reaction. However, the presence of its band in the IR spectrum excludes the possibility of the interaction of the metal ion with the amide group (NH-). A DFT calculation was used to interpret these results, which indicated an enolization process and keto-enol tautomerism of the complex. Computational studies were used to illustrate the optimized structures of the metal complexes and to understand alterations in NTZ upon complexation and its biological activity. EPR of the NTZ-Cu(II) compound revealed an octahedral arrangement around the metal ion. TGA degradation results agreed with the proposed molecular structures. UV-Vis and emission spectroscopy experiments indicated that Cu(II) and Ni(II) have stronger interactions with DNA molecules than NTZ-Zn(II), with the practical binding constant (K_b_) for the copper complex. HSA assays revealed a higher binding capacity of NTZ-Cu(II), but the zinc complex was the best candidate drug for release to the disease targets. An in-silico investigation with MOE software revealed the binding of NTZ to the active Cys145 site in the SARS-CoV-2 protease (6LU7). The divalent copper and nickel complexes potentially enhance the free ligand antiviral potency by bestowing better binding energies. Different in vitro approaches were used to test the antioxidant, antibacterial, and anticancer potency of the newly created complexes. NTZ-Ni(II) had the strongest antioxidant and bacterial inhibition activities, while NTZ-Cu(II) showed a practical effect against MCF-7 and HeLa with good selectivity. In summary, metal ions can impart different properties into compounds, depending on the metal. 

## Data Availability

All data generated or analysed during this study are included in this published article [and its Appendix A].

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
