# Peer review of "Investigating the Biological Potency of Nitazoxanide-Based Cu(II), Ni(II) and Zn(II) Complexes Synthesis, Characterization and Anti-COVID-19, Antioxidant, Antibacterial and Anticancer Activities"

_molecules, 2023, doi:10.3390/molecules28166126_

Round 1

Reviewer 1 Report

The manuscript shows potential for publication, but there are certain issues that need to be addressed by the authors before it can be accepted.

1. The abstract is concise and comprehensive, but it would benefit from the addition of a brief conclusion to complete the abstract.

2. To enhance the introduction and provide visual support, it is recommended to consider incorporating relevant figures, photos, maps, or diagrams.

3. The problem statement and novelty of the study should be stated with more clarity.

4. It is recommended to conduct a more critical analysis of previous studies. This would help establish a clear problem statement, identify the research gap, and determine the approach to fill it.

5. The methodology section is commendable. It is suggested to include a flowchart of methodology to aid in comprehending the methodology more effectively.

6. Improvements are needed in the quality of figures for Figure 1, Figure 3, Table 1 and Table 4 for better visual presentation.

7. Overall more in-depth discussion is necessary, especially to provide clarification and justification for the results. Certain sections of the current discussion are too concise and require expansion especially the analysis on antioxidant assay, antibacterial activity and In vitro antiproliferative activity.

8. It is recommended to provide a more in-depth and rigorous explanation of the underlying reasons behind the results, with a specific focus on addressing the question of "why does it happen?". For instance, the reason for the decrease in adsorption of both Cu(II) and Ni(II) complexes, while Zn(II) showed an increase has not been stated and should be addressed.

9. The conclusion is well written.

10. It is important to balance the length of paragraphs between being neither too short nor too long. This is to maintain readability throughout the text.

11. References are current and up-to-date.

The quality of English is satisfactorily good

Author Response

Dear reviewer

Please find attached our revised manuscript (molecules-2514244) entitled “Investigating the Biological Potency of Nitazoxanide-Based Cu(II), Ni(II) and Zn(II) Complexes Synthesis, Characterization, and Anti-COVID-19, Antioxidant, Antibacterial and Anticancer Activities”.

 We thank the reviewers for their insight and direction and feel that the manuscript has been significantly improved due to their input. We have addressed the reviewers’ comments and provide the revised manuscript along with detailed responses regarding the corrections (see below).

Reviewer 1

  1. the abstract is concise and comprehensive, but it would benefit from the addition of a brief conclusion to complete the abstract.

Thanks for this comment and a paragraph was added at the end.

  1. To enhance the introduction and provide visual support, it is recommended to consider incorporating relevant figures, photos, maps, or diagrams.

we agree with the reviewer but we want to focus on the results figure and table and don't distract the reader. However, a scheme was added.

  1. The problem statement and novelty of the study should be stated with more clarity.

A paragraph has been added for clarity.

  1. It is recommended to conduct a more critical analysis of previous studies. This would help establish a clear problem statement, identify the research gap, and determine the approach to fill it.

This was mentioned in the introduction line 59-69

  1. The methodology section is commendable. It is suggested to include a flowchartof methodology to aid in comprehending the methodology more effectively.

Thanks for this compliment. A scheme has added as requested.

  1. Improvements are needed in the quality of figures for Figure 1, Figure 3, Table 1 and Table 4 for better visual presentation.

Thanks for this comment. We tried our best to fix the figures and table.

  1. Overall more in-depth discussion is necessary, especially to provide clarification and justification for the results. Certain sections of the current discussion are too concise and require expansion especially the analysis on antioxidant assay, antibacterial activity and In vitro antiproliferative activity.

Thanks, more details were enhanced in the paragraphs.

  1. It is recommended to provide a more in-depth and rigorous explanation of the underlying reasons behind the results, with a specific focus on addressing the question of "why does it happen?". For instance, the reason for the decrease in adsorption of both Cu(II) and Ni(II) complexes, while Zn(II) showed an increase has not been stated and should be addressed.

Thanks for this comment. Actually, the three-transition metal are close in their prosperities and thus we perform lab and theoretical experiments to investigate which one has the best potency. The collected results give an indicator for the behavior of the compound in the biological system for pre-evolution for the next steps. The most important observation in the absorption and fluorescence experiments is the alternation of the peaks or the absorption intensity according to the addition of the compound. The answer to this question needs more advanced experiments. 

  1. The conclusion is well written.

Many thanks for this compliment.

  1. It is important to balance the length of paragraphs between being neither too short nor too long. This is to maintain readability throughout the text.

You are right, so we've edited some paragraphs to be balanced.

  1. References are current and up-to-date.

Many thanks for this comment.

Reviewer 2 Report

Review Report

Investigating the Biological Potency of Nitazoxanide-Based Cu(II), Ni(II) and Zn(II) Complexes Synthesis, Characterization, and Anti-COVID-19, Antioxidant, Antibacterial and Anticancer Activities

Overall, the research paper presents a valuable study on the synthesis, characterization, and biological activities of nitazoxanide-based metal complexes. However, it is evident that the paper requires careful revision to address various issues, such as clarification of basis set selection, correction of terminology, explanation of file formats and stable configuration determination. More importantly all the experimental and theoretical discussion of results must be separated under separate headings as in its current form, it is quite impossible to understand its contents.

Further clarification is needed regarding the use of different basis sets for different compounds.

Analytical and spectral detail of ligands is missing and must be provided in experimental

Why authors used these metals? Explain with reference to their importance in biology and medicine.

The terminology used in the paper, such as "absolute electronegativity" and "absolute hardness," should be corrected to "global electronegativity" and "global hardness" for accuracy.

The authors should provide an explanation for using a docking PDB file instead of a PDF file and its relevance to the research.

The methods employed to obtain stable configurations and select appropriate basis sets should be clearly outlined to ensure reproducibility.

Table 1 lacks clarity, particularly regarding the energies presented, and the range mentioned (-5767509.4 kJ/mol) requires further explanation or support from literature.

No conductance and magnetic data is provided? Geometry of complexes is not explained?

Table 3 needs clarification regarding the meaning of "pi" and should be titled as "global chemical" instead of "quantum chemical."

During the DFT modeling of complexes, the bond parameters before and after complexation must be discussed for reference see and cite Biometals 35 (3), 519-548

The calculation of Stokes shift changes from absorbance/emission spectra should be included for a more comprehensive analysis.

Provide the comparison from literature where necessary for this see Journal of Molecular Structure Volume 1254, 15 April 2022, 132305 and Journal of Enzyme Inhibition and Medicinal Chemistry Volume 28, 2013 - Issue 5

The absence of a binding energy table for molecular docking is a notable omission and should be addressed to provide insights into the interactions between metal complexes and target molecules.

Language polishing is required for whole manuscript

English language editing is required 

Author Response

Dear reviewer ,

Please find attached our revised manuscript (molecules-2514244) entitled “Investigating the Biological Potency of Nitazoxanide-Based Cu(II), Ni(II) and Zn(II) Complexes Synthesis, Characterization, and Anti-COVID-19, Antioxidant, Antibacterial and Anticancer Activities”.

 We thank the reviewers for their insight and direction and feel that the manuscript has been significantly improved due to their input. We have addressed the reviewers’ comments and provided the revised manuscript along with detailed responses regarding the corrections (see below).

Further clarification is needed regarding the use of different basis sets for different compounds.

Thank you for this comment, this corrected, and we add more detail.

Analytical and spectral detail of ligands is missing and must be provided in experimental.

The ligand is well known analytically and physically. Some data were shown for a comparison to find the alteration in the molecule and prove the structure as IR, NMR and theoretical calculation. Please check this reference for the complete characteristic properties for the NTZ ligand doi:10.1016/j.molstruc.2010.09.006.

Why authors used these metals? Explain with reference to their importance in biology and medicine.

Thank you for this comment, and we add more detail to the introduction.

The terminology used in the paper, such as "absolute electronegativity" and "absolute hardness," should be corrected to "global electronegativity" and "global hardness" for accuracy.

Thank you for this comment, this is corrected.

The authors should provide an explanation for using a docking PDB file instead of a PDF file and its relevance to the research.

Thank you for this comment, the MOE software just accepts this kind of file and it is well known for the MOE users. 

The methods employed to obtain stable configurations and select appropriate basis sets should be clearly outlined to ensure reproducibility.

Thank you for this comment, some additions were added.

Table 1 lacks clarity, particularly regarding the energies presented, and the range mentioned (-5767509.4 kJ/mol) requires further explanation or support from the literature.

These values were extracted from the theoretical calculation and dese not depend on other structures, we presented to find the most stable structure in the investigated media by comparing the suggested structures which less value show more stability.

No conductance and magnetic data is provided? Geometry of complexes is not explained?

These details were presented under the title ((General Chemical and physical characterization line 228, 282 while complexes geometry was discussed under Computational Studies line 315 to 324.

Table 3 needs clarification regarding the meaning of "pi" and should be titled as "global chemical" instead of "quantum chemical."

Thank you for this comment, this has been corrected.

During the DFT modeling of complexes, the bond parameters before and after complexation must be discussed for reference see and cite Biometals 35 (3), 519-548

Thank you for this comment, a sentence was added.

The calculation of Stokes shift changes from absorbance/emission spectra should be included for a more comprehensive analysis.

Thank you for this comment, it is interesting. The importance of Stokes shift is for practical applications of fluorescence of new compounds and allows to separate (strong) excitation light from (weak) emitted fluorescence. In our study, HSA is a well-known biological molecule and the aim is to investigate the binding ability of the new complexes to albumin as the first scan for further evaluation. 

The absence of a binding energy table for molecular docking is a notable omission and should be addressed to provide insights into the interactions between metal complexes and target molecules.

Thank you for this comment, the values were discussed and presented in the manuscript through the results and discussion. However, a table was added to support data and cited.

Reviewer 3 Report

I think the manuscript is interesting and helpful in the current practice of treatment for COVID-19. I have some comments as follows:

1. Many advances have been made in development of anti-SARS-CoV-2 inhibitor Nitazoxanide. For example, “Nitazoxanide (NTZ; Alinia®), an oral antiparasitic orphan drug developed by Romark Laboratories in the early 1970s, was the first, and currently the only, first‐line agent approved by the U.S. FDA for the treatment of Cryptosporidium or Giardia infections. Furthermore, NTZ has been found to exert anti‐SARS‐CoV‐2 effects in Vero E6 cells (EC50 of 2.12 μM) with moderate cytotoxicity (SI = 16.8). The combination of NTZ (0.6–5 μM) with other agents, such as remdesivir, amodiaquine, or umifenovir, produces significant synergistic effects compared with NTZ alone in Vero E6 cells. (DOI: 10.1002/jmv.27517)”

2. There is a lack of recent literature citations. For example, in lines 38-41, “Strategies to overcome these complications include drug encapsulation in nanocarriers, the development of prodrugs, and structural modifications by the addition of active groups or hetero cycles. (DOI: 10.1016/j.addr.2021.01.002; doi: 10.3390/biomedicines9060689; DOI: 10.1038/s41467-021-26760-4)”.

3. As for the anti-SARS-COV-2 potential. Please provide IC50 or EC50, please complete selectivity indexes for all compounds.

Minor editing of English language required

Author Response

Dear reviewer,

Please find attached our revised manuscript (molecules-2514244) entitled “Investigating the Biological Potency of Nitazoxanide-Based Cu(II), Ni(II) and Zn(II) Complexes Synthesis, Characterization, and Anti-COVID-19, Antioxidant, Antibacterial and Anticancer Activities”.

 1- We thank the reviewers for their insight and direction and feel that the manuscript has been significantly improved due to their input. We have addressed the reviewers’ comments and provide the revised manuscript along with detailed responses regarding the corrections (see below).

Many advances have been made in development of anti-SARS-CoV-2 inhibitor Nitazoxanide. For example, “Nitazoxanide (NTZ; Alinia®), an oral antiparasitic orphan drug developed by Romark Laboratories in the early 1970s, was the first, and currently the only, first‐line agent approved by the U.S. FDA for the treatment of Cryptosporidium or Giardia infections. Furthermore, NTZ has been found to exert anti‐SARS‐CoV‐2 effects in Vero E6 cells (EC50 of 2.12 μM) with moderate cytotoxicity (SI = 16.8). The combination of NTZ (0.6–5 μM) with other agents, such as remdesivir, amodiaquine, or umifenovir, produces significant synergistic effects compared with NTZ alone in Vero E6 cells. (DOI: 10.1002/jmv.27517)”

Thank you, I understood from the above paragraph that you want to compare the development of anti-SARS-CoV-2 inhibitor and you mentioned one method (combination with other approved drugs). Here we tried to enhance this potency with coordination with bio-metals ions. This aim was presented in the introduction.

  1. There is a lack of recent literature citations. For example, in lines 38-41, “Strategies to overcome these complications include drug encapsulation in nanocarriers, the development of prodrugs, and structural modifications by the addition of active groups or hetero cycles. (DOI: 10.1016/j.addr.2021.01.002; doi: 10.3390/biomedicines9060689; DOI: 10.1038/s41467-021-26760-4)”.
  2. As for the anti-SARS-COV-2 potential. Please provide IC50 or EC50, please complete selectivity indexes for all compounds.

Thank you for this comment. the investigation was carried out using a theoretical approach (MOE software). Providing the values of IC50 or EC50 can be extracted with in vitro or in vivo experiments which we hope our lab provided in the future.